# Associations between Parent Attitudes and on- and off-Screen Behaviours of Preschool Children in Singapore

**DOI:** 10.3390/ijerph191811508

**Published:** 2022-09-13

**Authors:** Michael Yong Hwa Chia, John Komar, Terence Buan Kiong Chua, Lee Yong Tay

**Affiliations:** 1Physical Education and Sports Science Academic Department, National Institute of Education, Nanyang Technological University, 1 Nanyang Walk, Singapore 637616, Singapore; 2Office of Education Research, National Institute of Education, Nanyang Technological University, 1 Nanyang Walk, Singapore 637616, Singapore

**Keywords:** early childhood, physical activity, sleep, media use, parenting

## Abstract

The research aims were to examine the relationships between parent and child digital media use and to describe the characteristics of the top and bottom quartiles of child digital media use in meeting the 24 h integrated guidelines for sleep, physical activity, and sedentary behaviour. Parent-reported on- and off-screen habits and quality of life of children were collected from 1481 parents of preschool children aged 2–5 years in 2019. Significant relationships were detected between parent and child digital media use (weekday, *r* = 0.274; weekend, *r* = 0.421, *p* < 0.05) with no sex difference in daily child digital media use (*p* > 0.05). Age of first exposure to fixed screens, daily digital media use, and physical play durations were significantly different between heavy (child-TQ) and light (child-BQ) child users of digital media (*p* < 0.05). Parental perceived importance of child digital media use and parental awareness and practice of guidelines for child digital media use differed significantly between parents of child-TQs and parents of child-BQs (*p* < 0.05). The number of 24 h movement guidelines met between child-TQs and child-BQs differed (*p* < 0.05). Parents voiced serious concerns for child digital media use, including addiction (75–76%), poor eyesight (73%), access to inappropriate content (73–74%), a lack of parent–child interaction (65%), poor sleep (49–55%), and a lack of physical activity (55–59%), but these findings did not translate to parental enforcement of the guidelines. Parent education on how to get the best digital media use outcomes for preschool children is recommended.

## 1. Introduction

The formative years of early childhood play a fundamental role in the development of health-related behaviours and are where family routines are amenable to change. The early childhood years are a time for critical brain development, a time for building secure relationships, and a time for establishing healthy behaviours. To ensure the best developmental outcomes for children, it is important to monitor, understand, and, perhaps, intervene where necessary, in the daily habits of children. Researchers posit that parental mediation of digital media use is associated with better outcomes for children. For example, active parent mediation (e.g., when parents discuss characters or themes from a television programme with the intent of promoting children’s critical-thinking skills) mitigates the risks of exposure to violent media, which can lead to aggression in children. Additionally, parental mediation of prosocial digital media use can also enhance prosocial behaviour in children. [1,2]. However, research into parent attitudes and mediation, especially on child use of mobile digital media devices, is in its relative infancy and is in need of research attention.

### 1.1. Pervasiveness of Digital Media Use in Early Childhood

Mobile phone access in Singapore is pervasive. Market research showed that the number of smartphone users in Singapore was estimated at 4.65 million in 2020, constituting more than 90% of the country’s total population [3] with 88.4% of its population having internet access [4]. As mobile digital media devices (smartphones and tablets) are common items among families in Singapore, it is conceivable that many of these adult-owned devices are used by young children either at home or elsewhere. How children use digital media devices is instructive. Data from the Singapore Longitudinal Development Study show that for children aged below 3 years, 33% used these electronic devices (computer, smartphone, or tablet) for learning at least once a week, but almost 50% used them for entertainment purposes (watching TV, movies, or music videos) at least once a week [5]. In this age group, 7.7% used the smartphone or tablet alone and about 50% used the device with an adult. In the same study, for children older than 3 but less than 6 years, about 60% used the devices for learning and games while 80.3% used them for TV shows, movies, and music at least once a week. One in four used the devices alone and 60% used the device with an adult [5]. In another study called the Growing Up in Singapore Towards healthy Outcomes (GUSTO) study, researchers reported that at age 2 years, children’s daily digital media use was 1.6 h and 0.7 h for television and mobile devices, respectively. At 3 years old, time spent on mobile devices increased by 0.3 h [6].

The increase in children’s use of mobile digital media devices over time is also seen in other countries. Between 2015 and 2019 in the UK, TV viewing among 3- and 4-year-old children decreased from 98 to 95%, while programmes viewed on mobile devices such as tablets and smart phones increased from 20 to 35% and 9 to 16%, respectively. Moreover, about one in four children owned a tablet [7]. In South Korea, ownership of tablets and time spent on digital media devices such as television, tablets, and smart phones increased significantly from 2015 to 2017 in the same group of preschool children followed from ages 2 to 5 years (TV by 0.6–0.7 h/day, tablet PCs by 0.6–0.8 h/day, and smartphones by 0.4 h/day) [8]. It appears that the pervasiveness of digital media use in early childhood is not a situation unique to Singapore as these reports collectively demonstrated that digital media access and engagement among preschool children in developed countries across regions are ubiquitous and are increasing over time and with an increase in the age of children.

### 1.2. Benefits and Risks Associated with Early Digital Media Use

A review of 30 papers between 2009 and 2014 provides an insightful account of the benefits and demerits of digital media use in children aged 3–6 years old in school settings. The review highlighted many methodological differences across studies, making it difficult to generalise findings, but it was able to highlight benefits of technology use in school settings for literacy, engagement, social interactions, and mathematics [9]. The review covered studies that reported on traditional forms of digital media use such as television viewing and desktop computers but less on more recent forms of mobile digital media use with smartphones and tablets. These mobile devices are now omnipresent and almost inescapable in the lives of young children.

Many studies show educational benefits for young people engaged in digital media use. These educational benefits were shown to be higher in healthy children than in children at risk, albeit this was reported in older children. Nonetheless, the effect sizes are educationally meaningful for both cohorts of children [10]. Other scholars opine that an early adoption of digital media predisposes preschool children to physically inactive indoor lifestyles [11,12], displaces outdoor play [13], increases the risk of an earlier development of myopia [14], affects sleep duration and sleep quality [15], impairs self-regulation [16], and reduces the quality of parent–child interactions [17], among other insidious outcomes [18]. However, it appears that the jury is still out in terms of whether the use of digital media among children aged 2–5 years is beneficial or not. In the USA, 31% of parents of children under the age of 12 years shared that they were very concerned that children already spend too much time engaged in digital media devices and that parenting is harder than it was 20 years ago with parents citing technologies such as social media and smartphones as reasons for concern [19].

The scientific consensus is that addictive behaviours generally arise from a combination of an individual’s genetic predisposition and environmental risk factors that promote exposure to a specific substrate [20]. In the context of digital media addiction in children, a deeper understanding of the quantity and quality of the digital media exposure is required to better discern the underlying antecedents and reasons for the addiction [21]. For instance, while gaming disorder is a diagnosable condition in older children, social media and non-gaming addictive behaviours are not addressed. Nonetheless, ‘digital addiction’ represents a real and potential problem that defies easy solutions or prevention strategies [21]. More research is required in discerning the addictive potential of excessive digital media exposure in early childhood.

### 1.3. Screen-Media- and Movement-Associated Guidelines for Young Children

In response to the increasing use of digital media, international (e.g., American Academy of Pediatrics) and national (e.g., Early Childhood Development Agency) associations published guidelines on how preschool children should use digital media in school and in general (e.g., Media Literacy Council). While some guidelines vary slightly, such as age group cut-offs (e.g., less than 2 years, 3–5 years) to suit the different contexts of the countries, there is a universality of principles in the recommendations for healthy screen engagement by young children for parents i.e., limit daily digital media use to no more than 60 minutes, select only age-appropriate content for viewing, co-view or co-engage with children when engaged when using digital media, schedule off-screen activities with children and set up digital media-free zones (e.g., in the bedroom) and times (e.g., at meal times or when outdoors).These guidelines and parenting tips for managing digital media use in children are anchored on the premise that children learn best when they are physically engaged in real-time in-person interactions with peers or with a significant adult (parent, teacher, or caregiver). 

Guidelines on Physical Activity, Sedentary Behaviour and Sleep for Children under 5 years of age became available in 2019. These 24-hour integrated guidelines, promoted by the World Health Organization, stipulate that children between 2 and 5 years should, on a daily basis, have 10–13 hours of good quality sleep, 3 hours of physical activity (inclusive of 1 hour of activities that are of moderate-to-vigorous intensity), not be restrained and kept sedentary for more than 1 hour at a stretch, and accumulate no more than 1 hour of screen time, where less is better [22]. As screen time is usually engaged in while sitting, this activity is considered as a constituent of sedentary behaviour (i.e., sitting, reclining, or lying outside of sleeping), albeit sedentary behaviour also includes off-screen activities such as reading, colouring, doing puzzles, chatting on the phone, and playing board games. Meeting or not meeting these movement guidelines may affect significant outcomes in the quality of life of children. A study in Singapore showed that health-related quality of life scores of preschool children who met all three 24-hour movement guidelines were significantly higher than those of children who met fewer of the guidelines [23]. There are merits in the continued monitoring of lifestyle behaviours of children in childhood. Surveillance of on-screen and off-screen behaviours among preschool children in Singapore provides useful information for early intervention and ameliorative programmes to address gaps in the holistic development of preschool children so that they are well-prepared for more formal education in later childhood and adolescence. 

### 1.4. Importance of Parent Modelling

As parents are children’s ‘first teachers’ and caregivers and children’s most important role models, they must be empowered and supported in ways that can bring out the best outcomes of screen media engagement in preschool children. Understanding parents’ use of digital media, their attitudes, concerns, and behaviours about their children’s use of technology in a highly internet-connected society such as Singapore provides evidence-based and context-relevant data for parents to work with teachers in harnessing the benefits and mitigating the risks of preschool children’s engagement with technology. These data are also ‘rich fodder’ for further research. Researchers in an experimental study reported that the quality of parent–child interactions was negatively affected when parents were engaged in smartphone use (intervention condition) compared to when parents did not use the smartphone (control condition) when spending time with children who were 5 years old [17]. The study provided evidence that parents who used their smartphones when spending time with children felt more distracted and reported lower feelings of social connection in general, and specifically with children. Another study showed that mothers’ and fathers’ technology use disrupted parent–child interactions, resulting in sub-optimal interactions and was associated with child behavioural problems in 3-year-old children [16]. However, researchers of another study reported that Greek parents had positive attitudes towards their children’s use of mobile technologies and wanted to support their learning by providing a stimulating learning environment at home [24].

Past research on TV viewing (i.e., a form of screen media) shows that parent mediation—how parents interact or not with children around screen media—may mitigate the negative effects and enhance the positive effects of children’s screen media use [1]. By extrapolation, parent mediation in children’s use of mobile digital media is a fertile ground for research.

Given that a child’s behaviour is largely influenced by his or her parents, it is important to investigate the relationships between on-screen and off-screen behaviours of preschool children and the attitudes of their parents toward these behaviors of children. These data provide ‘just-in-time’ and ‘current-state’ evidence to inform parents, policies, and processes that enhance the quality of early childhood education at home in partnership with teachers in school. In addition, findings of the present study will inform parents especially about the importance of optimising time use in digital media, physical activity, and sleep in order to maximise the best outcomes of the health and wellbeing of their children.

### 1.5. Study Purpose

The purpose of the present study was two-fold: (i) to examine the parent–child nexus in terms of on-screen and off-screen behaviours of preschool children, and (ii) to compare the characteristics of the top and bottom quartiles of children’s daily digital media use. These characteristics include the home digital media environment, parental use of digital media, parental attitudes, concerns, awareness and enforcement of child digital media use guidelines, age of child’s first exposure to digital media, child’s daily on-screen and off-screen behaviours (indoor and outdoor play), demographic (parent education and household income) and child-related information (sleep and eyesight), and the proportion of children meeting all three or meeting none of the 24-hour integrated guidelines. The usefulness of comparisons between the characteristics of the top 25% (child-TQ) versus bottom 25% (child-BQ) quartiles of daily child digital media use allow for benchmarking against the recommendations of the ‘ideal’ state or situation and for process improvements [25].

Specifically, the research questions were:What is the relationship between parent digital media use and child digital media use on a weekday and a weekend day?Are there differences in age of first exposure to fixed and mobile screens, daily media use, physical play, sleep, and quality of life between heavy child users (child-TQ) and light child users of digital media (child-BQ), respectively?Are there differences in digital media habits, attitudes toward child digital media use, knowledge, and enforcement of child digital media guidelines between parents of child-TQs and parents of child-BQs?Do child-TQs and child-BQs meet the WHO 24-hour Guidelines on Physical Activity, Sedentary Behavior and Sleep?

## 2. Materials and Methods

### 2.1. Ethics Clearance and Participant Recruitment

Ethics clearance for the study was granted by the university institutional review board (NTU IRB-2017-09-036). Parents of children attending 425 preschools in Singapore (a tropical urbanised city state, situated 1.5 degrees north of the Equator) in 2019 between the months of March and August were recruited. These preschools operated under a private–public enterprise arrangement called the Anchor Operator Scheme [26]. The private–public arrangement provided government-subsidised and good quality early childhood education to the masses, and especially to lower income and disadvantaged households. Our data showed that the majority of parents who enrolled their children under the Anchor Operator Scheme came from lower-to-middle-income households (Table 1). The preschool operators sent out a research invitation letter inviting parents of children enrolled in their preschools to take part in the survey. Inclusion criteria were parents with children aged 2–5 years. Parents provided informed consent to their voluntary participation and were assured of anonymity of their survey responses.

### 2.2. Survey Instruments and Mode of Data Acquisition

An online questionnaire package consisting of the Surveillance of Digital Media Habits in Early Childhood Questionnaire (SMALLQ^®^) and the Pediatric Quality of Life 4.0 Inventory (PedsQL™ 4.0) was hosted on a secured survey platform called Qualtrics. Both SMALLQ^®^ and PedsQL™ 4.0 have been previously used together in Singapore in different groups of parents of preschool-aged child [23].

#### 2.2.1. Surveillance of Digital Media Habits in Early Childhood Questionnaire (SMALLQ^®^)

The SMALLQ^®^ comprised three broad segments: (a) digital media use by parent and child; (b) child off-screen activities; (c) demographical characteristics of parent and child. The SMALLQ^®^ is not a psychometric instrument tool. Rather, the questionnaire was developed to solicit from adult parent information about the digital media (screen time, type, and purpose) and non-digital media (indoor and outdoor play, sleep duration, and sleep quality) habits of their children on a weekday and on the weekend. Parents were asked about their attitudes toward child digital media use, and their awareness and practice of child digital media use guidelines [27].

The development of SMALLQ^®^ was guided by the AMEE framework—Guide No.87 [28]. The 7-step framework was advocated to develop high-quality surveys that are suitable for research and programme evaluation. These seven steps include (1) literature review, (2) focus group interviews, (3) synthesis of the literature review and focus group outcomes, (4) development, modification, or re-order of the questionnaire items, (5) feedback on the items through expert opinion, (6) employment of cognitive interviews to ensure that respondents understand the items as intended, and (7) pilot-testing the online questionnaire. Questions on digital media use in SMALLQ^®^ were framed in the domains of content (e.g., is the digital media use for education or for entertainment), context (e.g., whether the parent is present with the child when digital media is in use; how digital media is delivered—computer, smart phone, game console), and dose (e.g., parent digital media use duration and child digital media use duration This was the conceptual understanding of how parents may influence their child’s digital media behaviour proposed by a group of screen media and parenting researchers [29]. In the development of the SMALLQ®, five parents with children aged below five years old, who were not involved in the research provided input on all items in the draft questionnaire and on the cognitive load of the questionnaire. Based upon their feedback, some questionnaire items were re-worded or re-ordered, thereby front-loading the validity of the SMALLQ^®^ [29]. The items in SMALLQ^®^ were then face- and content-validated by four independent experts (two academics with PhDs and two early-childhood educators with master’s degrees) based upon the criteria of representativeness, clarity, and relevance [28]. The development and validity of the SMALLQ^®^ questionnaire were previously described by the authors [27].

The internal consistency or reliability of the SMALLQ^®^ was also established. Based upon parent-reported child habits for physical activity (PA), child and parent habits for digital media (screen time, ST), and sleep (SL) on a weekday and on a weekend day, the Cronbach values were 0.78 for PA, 0.74 for ST 0.74, and 0.69 for SL. A Cronbach’s alpha of 0.70 is considered to have acceptable reliability for exploratory research such as the present study [30]. Overall, the online SMALLQ^®^ was judged to have adequate face and content validity with acceptable internal consistency or reliability for use in children in early childhood.

#### 2.2.2. Pediatric Quality of Life 4.0 Inventory (PedsQL™ 4.0)

The PedsQL™ 4.0 was used to measure the health-related quality of life of preschool-aged children. The 21- or 23-item questionnaire (21 items for parents of a preschool child aged 2 to 4 years and 23 items for parents of a preschool child aged 5–6 years) yielded a composite health score from scores of four dimensions of quality of life of a child (i.e., physical functioning, emotional functioning, social functioning, and school functioning). The psychometric properties of PedsQL™ 4.0 were studied by researchers, and the validity and reliability of the survey instrument are well-established [31,32].

### 2.3. Data Capture and Processing

The online questionnaire package was rolled out across five major preschool operators and kept ‘active’ for up to two months (shorter periods for some preschools). Parents gave their informed consent to participate before filling out the online survey. The survey was hosted on the Qualtric online survey platform (SAP, Provo, Utah), and data were captured at the backend. Reminders to complete the questionnaire were sent to parents by the preschools. This approach was adapted and adopted using the Total Design Method to enhance parent participation [33]. Responses to each of the online survey questions were not mandatory to avoid any form of coercion and reduce any form of social desirability bias. The number of completed responses was recorded.

The online questionnaire was deactivated at the end of data collection, and a copy of the raw dataset was downloaded from Qualtrics^®XM^ (SAP, Provo, Utah). Incomplete answers (e.g., when the responder started the survey but did not respond to any question) were checked, with a cut-off completion rate set at 90% of the total number of questions (i.e., respondents with less than 90% of the questions completed were removed). For each question, the distribution of the responses was visually checked, and any response that was outside the range of possibilities was removed (i.e., any outlier or seemingly impossible answer).

### 2.4. Statistical Analyses

#### 2.4.1. Parent-Reported Time Spent on Screen Media 

The daily digital media use of the child was derived from averaging the total digital media use (i.e., using screen media for education, using screen media for entertainment, creating media, and using screen media for communicating) on a weekday and a weekend day. Parent self-reported digital media habits (i.e., using screen media for work, using screen media for entertainment, using screen media for social networking, using screen media for personal development) on a weekday and a weekend day were summed and averaged to derive the daily parent digital media use. 

#### 2.4.2. Top and Bottom Quartiles of Child Screen Media Use

Quartile statistics possess a great deal of analytical power and are a simple yet useful technique for benchmarking purposes. In the context of the present study, quartiles (Qs) anchored upon parent-reported total child digital media use per day were generated. The characteristics of the top (child-TQ) and bottom 25% (child-BQ) (child-TQ and child-BQ as independent variables) of total child media use (as dependent variable) were foregrounded to allow for comparisons [25] in time use for physical activity (PA), screen time (ST), and sleep (SL) in children between 2 and 5 years of age. The use of quartiles allowed for better differentiation and clearer comparison between groups. These time-use behaviours were then benchmarked against the WHO 24-hour integrated guidelines for PA, ST, and SL for children aged under 5 years [22].

#### 2.4.3. Benchmarking against 24-Hour Guidelines on Physical Activity, Sedentary Behaviour, and Sleep

Descriptive statistics for the proportion of children who met and did not meet the physical activity, screen time, and sleep guidelines were generated—*meeting all* three guidelines, and *meeting none* of the 24 h integrated WHO guidelines for physical activity, sedentary behaviour and sleep for children aged between 2 and 5 years. Differences between the top and bottom quartiles in child digital media use (independent variable) for (a) the duration of physical play, sedentary activities and sleep (dependent variables), and (b) quality-of-life scores (dependent variable) were examined using independent-sample t-tests. The measure of effect size was reported as Cohen’s *d*, where *d* = 0.2 is interpreted as small effect size, *d* = 0.5 is interpreted as medium effect size, and *d* = 0.8 is interpreted as large effect size [34].

#### 2.4.4. Relationship between Parent and Child Digital Media Use

Pearson correlations between parent and child use of digital media were generated to examine and establish the parent–child digital media use nexus. When the Pearson test was significant, a value of *r* between 0.1 and 0.3 was interpreted as a small correlation, 0.3–0.5 was interpreted as a moderate correlation, and above 0.5 was interpreted as a large correlation [34].

#### 2.4.5. Non-Parametric Tests

When assumptions of normality were violated, the Mann–Whitney test was used as a non-parametric alternative. Effect size was reported as a Glass rank bi-serial correlation coefficient [35]. To examine the distribution of categorical answers between top and bottom quartiles of child digital media use (i.e., number of WHO 24 h integrated guidelines met, parental concerns about child on-screen behaviour, parent perceptions about the importance of using digital media, parent awareness and practice of guidelines for child digital media use, and child sleep quality are dependent variables of interest), chi-square tests of independence were used. Statistical significance for the parametric and non-parametric tests was set at *p* < 0.05. Analyses were completed using SPSS Statistics Version 26.0 (IBM Corp., Armonk, NY, USA) and JASP Version 0.14 (https://jasp-stats.org; accessed on 24 August 2022), a free and open-source programme supported by the University of Amsterdam).

## 3. Results

### 3.1. Characteristics of Survey Respondents

The research collected 2196 valid responses (survey completion of more than 90%). Data from 1481 parents/legal guardians of children aged 2–5 years (cut-off at 4.9 years) were included in the data analysis to be in line with the 2019 WHO integrated 24 h guidelines on physical activity, sedentary behaviour, and sleep for children under 5 years of age. The remaining 715 completed surveys were parent-reported for children aged 5 years and above but less than 6 years old. These were excluded from the dataset in the present study (for ease of comparison in relation to the WHO 24-hour guidelines) but would be used in the on-going national surveillance research. Of the 1481 respondents, 78.5% were mothers, 21.1% were fathers, and 0.3% were legal guardians. The ethnic composition of the study sample (i.e., 70.9% Chinese, 13.5% Malays, 10.8% Indians, and 4.8% Others) closely reflected the country’s population profile [36].

### 3.2. Parent–Child Nexus in Digital Media Use and Characteristics of Child Digital Media Use

Pearson’s correlation analysis revealed that parent daily media use was significantly correlated to child daily media use on the weekday (*r* = 0.27, *p* = 0.001 *n* = 1481). The significant relationship increased from small to moderate on the weekend day (*r* = 0.42, *p* = 0.001, *n* = 1481).

### 3.3. First Exposure to Fixed and Mobile Screens, Daily Media Use, Physical Play, Sleep, and Quality of Life between Child-TQs and Child-BQs

Table 1 presents a summary of the quartile characteristic (top vs. bottom 25% of data) of daily child digital media and that of their parent digital media use, and the proportion of children meeting the 24-hour WHO integrated guidelines for SL, PA, and sedentary ST. The age of first exposure to fixed screens of child-TQs was significantly earlier than that of child-BQs (17.1 vs. 18.9 months, *p* = 0.005). The age of first exposure to mobile screens was not significantly different between child-TQs and child-BQs (20.6 vs. 21.9 months, *p*= 0.110). Daily digital media use of child-TQs was significantly higher than that of child-BQs (4.8 vs. 0.6 h, *p* = 0.001). Child-TQs spent 0.9 hours more engaging in daily indoor and outdoor playtime than child-BQs (3.6 vs. 2.7 hours, *p* = 0.001). Sleep duration and quality were not significantly different between child-TQs and child-BQs (*p* > 0.05). Quality-of-life scores between child-TQs and child-BQs were not significantly different (*p* > 0.05).

### 3.4. Adherence to WHO 24-Hour Movement Guidelines between Child-TQs and Child-BQs

A chi-square test of homogeneity was run and found that the two multinomial probability distributions were not equal in the population, χ^2^(3) = 211.61, *p* < 0.001. No children in the child-TQ group met all three guidelines, while 31.0% of the child-BQ met all three guidelines. A total of 12.1% of child-TQs did not meet any guidelines, while all in the child-BQs met at least one guideline. 

### 3.5. Attitudes toward Child Digital Media Use, Knowledge, and Enforcement of Child Digital Media Guidelines between Parents of Child-TQs and Parents of Child-BQs

Chi-square tests of homogeneity were run and found that parental perceptions to the different merits of child digital media use except for one (put child to sleep) were significantly different between parents of child-TQs and parents of child-BQs (*p* < 0.05) (Table 2). Parents of child-TQs and parents of child-BQs were not significantly different in any of the parental concerns (*p* > 0.05) (Table 2). The proportion of responses on the awareness and enforcement of child digital media guidelines between parents of child-TQs and parents of child-BQs was significantly different (*p* < 0.05) (Table 3). Observed frequencies and percentages of parents’ perception, concerns, knowledge, and enforcement of child digital media guidelines for the top versus bottom quartiles are presented in Table 2 and Table 3.

### 3.6. Gender Difference—Weekday and Weekend Day in Child Digital Media Use

The proportion of female vs. male children reported was about equal (52.5% were boys, 47.5% were girls). No gender differences in parent-reported child media use on a weekday (2.0 ± 1.9 vs. 2.0 ± 1.9 h, *t*(1414) = −0.06, *p* = 0.956) or weekend day (2.8 ± 2.6 vs. 2.6 ± 2.2 h, *t*(1414) = 1.9, *p* = 0.058), or mean of both days combined (2.4 ± 2.1 vs. 2.3 ± 1.9 h, *t*(1414) = 1.12, *p* = 0.265) was detected. Child digital media use on the weekend was 1.3 to 1.4 times that of the weekday.

### 3.7. Home Digital Media Environment

No significant differences (*p* > 0.05) were found in terms of availability and access to smartphones and tablets (97.3 vs. 96.0%) and Blu-ray/DVD/CD/video tape players (17.3 vs. 21.6%), and technology toys (18.3 vs. 15.4%) for the child-TQs and child-BQs, respectively, for users of digital media. Contrarily, differences were detected for television (97.8 vs. 95.0%, *X*^2^(1, *n* = 807) = 4.36, *p* = 0.037), computers (64.4 vs. 74.4%, *X*^2^ (1, *n* = 807) = 9.66, *p* = 0.002), and video game devices (24.5 vs. 17.4%, *X*^2^ (1, *n* = 807) = 6.20, *p* = 0.013).

## 4. Discussion

The main purpose of study was to examine the relationship between parent’s daily time use of digital media and those of their children aged between 2 and 5 years old in a representative population sample in Singapore. This research is important as evidence elsewhere shows that younger children are gaining access and some have ownership of mobile technologies by age 4 years [37]. In the cited research, up to 75% of parents sampled reported that their children owned a mobile technological device. Such early adoption of digital media by children concurs with the present findings. In our study, the mean ages of first exposure to digital media devices (fixed vs. mobile screens) among preschool children were 18 and 21 months, respectively. Children in the child-TQ group were exposed to fixed screens such as television at a significantly earlier age than children in the child-BQ group. First exposure to mobile screens such as tablets and mobile phones by children in both child-TQ and child-BQ groups was before 2 years. Moreover, parent survey data on a nationally representative sample of 3640 adults living in the USA show that 66% reported that parenting is harder in the present time compared to 20 years ago with many citing that technologies such as smartphones and use of social media as a reason [19] (p. 3). A total of 71% believe that the widespread use of smartphone by young children might potentially result in more harm than good [19] (p. 4) and that 68% of parents reported that they felt distracted by their phone use when they spent time with their children [19] (p. 12). These data foreground the ubiquity of mobile technologies and the attention-consuming nature of digital media use and the range and complexities of parent perception and parent adequacy in dealing with digital screen media use in young people.

Emergent research show that parental mediation of screen media is associated with better outcomes for children and that parents have adopted different approaches in dealing with young children’s digital screen media use at home [1]. An example of parental mediation of screen media is active mediation (e.g., when parents discuss characters or themes from a television programme with the intent of promoting their child’s critical-thinking skills). Research shows that risks of negative effects of screen media use such as exposure to violent media can be mitigated as well as positive effects such as positive prosocial behaviour can be reinforced through active parental mediation [1,2]. As parent screen media use is a significant predictor of child digital media use [38], reducing parental digital media use when in the presence of children and enhancing parent–child interactions is an important area of behaviour change. It is important that parents are supported and equipped with the appropriate skills in managing the dynamic and complicated nature of child digital media use. 

The present research surveyed the digital and non-digital media habits of parents and their preschool-going-age (2–5 years) children in a nationally representative sample, thereby highlighting the importance of the present findings.

### 4.1. Parent–Child Nexus in Digital Media Use and Home Digital Media Environment

Significant and meaningful associations were established between daily parental digital media use and child daily digital media use on weekdays and this relationship increased in strength on weekend days. These results compare favourably with data reported in another study [39]. In the cited study of 2326 American parents with children aged between 0 and 8 years, children’s screen time on television, computers, smart phones, and tablets was strongly predicted by parents’ own use across these digital platforms. 

In the present study, parents with lower education, and those from households with lower incomes, were more represented in the top quartile (child-TQ, i.e., highest 25% of daily child digital media use), yet parent ownership of smart phones and tablets was not significantly different between the top and bottom quartiles of daily child digital media use (i.e., parent ownership of smart phones and tablets for both groups was at least 95%). These observations were affirmed by another study [37] where children from economically disadvantaged households, are using newer digital technologies, such as interactive and mobile media, on a daily basis, and these groups of children continue to be the target of intense marketing [18].

### 4.2. Digital Media Use on Weekday and Weekend 

Child digital media use in the weekend was 1.4 times that on a weekday. On weekdays, children attend preschools for between seven hours and twelve hours in Singapore, leaving less time for home and out-of-school activities compared to on the weekend [5]. In the present study, the majority of preschool children was enrolled in the 12-hour full-day programme. Parent digital media use was significantly lower on the weekend compared to on weekdays (range: 55–72% of weekday use; Table 1). This is an interesting observation in that while parents have a ‘digital respite or pause’ on the weekend, parents allowed children to indulge in a ‘digital feast’ (i.e., more screen time perhaps because there was no need to attend school) on the weekend. A plausible explanation is that on weekends, some parents devote time to non-work and non-screen-based activities such as running errands and doing household chores while providing children with more opportunities to access digital media, perhaps using digital media time as a ‘digital nanny’. These findings contrast with research on weekday-weekend parent–child screen time and physical activity behaviours in children aged 5–12 years in Czech families [40,41]. In the cited studies, parents who achieved physical activity guidelines on weekdays and weekend days (defined as achieving a daily step count of 10,000 steps or more) are more likely to have children also meeting the daily physical activity guidelines. Conversely, excessive weekend parental screen time reduced the likelihood of children achieving the recommended daily physical activity.

### 4.3. Characteristics of Children in the Top and Bottom Quartiles (Child-TQ and Child-BQ)

Another aim of the study was to compare the characteristics of children in the top and bottom quartiles of daily digital media use (i.e., child-TQ vs. child-BQ). Such comparisons are useful for benchmarking and for process evaluations [25], which in the present research, include benchmarking, against the World Health Organization’s 24-hour integrated guidelines on sleep, physical activity, and sedentary screen time for children below 5 years of age.

Considering that a child’s digital media habits are largely influenced by parents, comparing the characteristics of parents of children in the top and bottom quartiles in terms of their digital media habits, attitudes toward their child digital media use, and their awareness and practice of digital-media-associated guidelines offer illuminating insights and form the basis for future intervention to ameliorate health-depreciating lifestyle habits before they become entrenched.

### 4.4. Household Income and Education Levels

Parents clustered in the bottom quartile (lowest 25% in daily child digital media use; child-BQ) had these characteristics in comparison to those clustered in the top quartile (highest 25% daily in child digital media use; child-TQs): they came from households that were more affluent (greater than SGD 8000 per month), had higher academic qualifications (at least university qualifications), and 43% lower daily digital media use (see Table 1).

### 4.5. Parental Perceptions of Digital Media Use and Enforcement of Guidelines

When it came to reporting on their enforcement of a guideline, e.g., limiting screen time to an hour a day, parents of both the child-BQs and child-TQs differed significantly (see Table 3). In terms of perceptions about daily child digital media use, the parents in the child-BQ group were significantly less concerned than parents in the child-TQ about using digital media for improving children’s knowledge and skills, for keeping children entertained, for communications, for keeping children occupied, and as a tool to divert the child’s attention. However, neither child-TQ or child-BQ of parents differed significantly in their perceptions about using digital media for helping the child to sleep (see Table 2). No significant difference was flagged by parents of child-BQs and child-TQs for any of the parental concerns that were asked (poor sleep, poor eyesight, lack of physical activity and exercise, exposure to inappropriate content, addiction fears, and the lack of parent–child interaction) (see Table 2).

Additionally, parents of child-TQ tended not to follow digital media guidelines for children compared to parents of child-BQ, specifically for the following guidelines: introduce only highly educational content or programmes to children and co-viewing or co-playing digital media with children (see Table 3). It appeared that there was less parental supervision and involvement with child digital media activities in the child-TQ. Parents of child-TQ felt more strongly than parents in the child-BQ about the merits of digital media use for child acquiring skills and knowledge, for entertaining the child, for communications, for keeping the child occupied, and for diverting the child’s attention.

These parental perceptions and concerns and attitudes mirror the findings reported in another study [39] where child use of digital media appears to be the result of an interaction between child and parent factors and is highly influenced by parental attitudes [39]. Parental and child digital media use were moderately associated with each other, especially on the weekend day in the present study, supporting the findings of the cited authors [39].

### 4.6. Digital Media Use

From Table 1, daily digital screen media use in the child-TQ was 4.8 hours, and this was eight times that of children in the child-BQ. Even then, weekend digital media use for children in the child-TQ and child-BQ were 5.5 and 0.68 times, respectively, of the maximum 1-hour guideline for daily digital use for children in this age group. This excessive child digital media engagement over the weekend may be at the expense of other meaningful non-digital parent–child activities (such as spending time outdoors together and, engaging in parent–child physical play). The excessive daily ST may become entrenched as a habit and become normalized and could eventually result in poorer physical, cognitive, and developmental outcomes for children [18].

### 4.7. Quality of Life

Of interest is that parents in the child-TQ reported a lower total health score (i.e., poorer quality of life, QoL), and with none of the children meeting all three guidelines for sleep, physical activity, and sedentary behaviour compared to the number of guidelines met by children in the child-BQ. Overall, 31% of child-BQs met all the WHO 24-hour integrated guidelines while 12.1% of child-TQs did not meet any of the guidelines. Parents of the child-TQs spent a daily average of nine hours on digital media compared to five hours for parents of the child-BQs. It appears that the ancient proverb ‘like parent, like child’ in terms of on-screen behaviour is compelling since young children learn new skills and behaviours by imitating and modelling behaviours of their parents. For instance, a study examined the food selection of 120 children aged 2–6 years and showed that when presented with a wide array of food products, young children chose combinations of healthier and less healthy foods and beverages, and that children assimilated and mimiced their parents’ food choices at a very young age, even before they were able to fully appreciate the implications of their choices [42]. Research in older children show associations between children’s addictive behaviours such as alcoholism, smoking, and destructive behaviours, and parental harmful behaviours and habits [43]. These studies reinforce the importance of positive parental modelling (including practising healthy digital media habits) in the positive inculcation of lifestyle habits in preschool children.

### 4.8. Parental Disconnect between Concern and Action

From Table 3, while a sizeable percentage of parents of children in the child-TQ voiced concerns about the impact of digital media use from poor sleep (49%) to fear of addiction (76%), these concerns did not translate into positive action even when 32–60% of parents in the same quartile were aware and practising (or enforcing) the digital media use guidelines for children.

This disconnect between parental concern and a lack of ameliorative action is also highlighted by Pearson and his colleagues (2011) [44]. In the cited study, parents who were concerned about the impact of excessive TV viewing in children had reason to be but might not have been aware of certain parenting practices (e.g., excessive parent TV viewing habits) that influence child TV-viewing habits. The authors recommended family-based intervention with education and support elements to help parents make changes to the family environment to reduce child TV viewing as possible actions to take.

### 4.9. Strengths and Limitations

The present research presented useful data on the daily child PA, sedentary ST, and SL in a racially diverse and country-representative sample of preschool children in Singapore. The research alluded to the importance of parent modelling in forming wholesome habits in the daily use of digital media for preschool children. The research contrasted the characteristics of the top and bottom quartiles of child media users, especially in terms of the proportion of preschool children meeting the 24-hour integrated guidelines for SL, PA, and ST, and concomitantly, the QoL of children. The attitudes of parents of children from the top and bottom quartiles of child media users were compared. The differences in parent habits, concerns, and attitudes toward the digital media use of their children allow for targeted inventions to better support parents in need of help in managing family ST, PA, and SL. Importantly, these comparisons signal the need to re-balance the digital and non-digital habits of children in the top quartile (i.e. child-TQ).

A limitation of the research was that the research was dependent on the parent recall of habits and behaviours, but this was mitigated by using recall periods of no more than seven days (i.e., 7-day recall). A self-report of parent attitudes and whether parents practised digital media use guidelines might also be limited by social desirability bias [45]. This bias was, however, mitigated by front-loading the validity (i.e., involving parents with preschool children in co-developing the questionnaire) of the online questionnaire [27] and using the SMALLQ^®^—an anonymous questionnaire with no identifiers. All parent responses were also confidential and secured by institutional protocols (in compliance with the university’s data management policies). Second, given that the guidelines were relatively new when this research was undertaken, it is conceivable that knowledge of this guidelines have yet to filter down to parents, which might account for this lack of awareness. Therefore, efforts to publicise these guidelines should be sustained.

### 4.10. Way Forward

In terms of preschool children growing, developing, and living in a saturated screen media environment, child characteristics, the parent–child relationship, familial dynamics that include sibling interactions, parental mediation practices, and parents’ own use of media, singly or in combination, can influence children’s digital media use. More research is required to help explain the contributory roles that each or all, in combination, play in developing wholesome and well-adjusted children. Makers of policies should also consider the home or family environment to better address parental needs (e.g., single-parent families, working parents, multigenerational families living in the same household, children with special needs) to influence digital media use of preschool children for healthy development. 

In Singapore, such targeted approaches and early intervention for parents in need, especially those from lower-income households, require inter-ministerial (Ministry of Education, Ministry of Health, Ministry of Culture, Community and Youth, Ministry of Social and Family Affairs) coordination and cooperation. An example of such a scheme is KidSTART. Eligible parents under KidSTART are provided with an ‘ecosystem of support’ around the child. Multidisciplinary professionals and the community work together with parents in response to specific needs of the child and family (including digital media literacy and associated issues) with the aim of enabling children in specific circumstances to achieve age-appropriate development in his immediate settings (https://www.msf.gov.sg/media-room/Pages/Fact-sheet-on-KidSTART.aspx accessed on 24 August 2022) As robust resources are funnelled over a sustained period to support the KidSTART programme, research that tracks the progress of families when children enrolled between 0 and 3 years and follows up till just prior to transition to primary school at 6 years old is necessary to evaluate its effectiveness and holistic impact on children enrolled in the programme. In this regard, the Centre of Research in Child Development at the National Institute of Education (https://nie.edu.sg/research/research-offices/office-of-education-research/centre-for-research-in-child-development; accessed on 24 August 2022) is well-placed to undertake such research.

Additionally, stepped up and targeted efforts in educating parents in need of help (e.g., identified heavy users of technology) on how to obtain the best digital media returns in early childhood and enabling support through ‘parents supporting parents’-type networks or access to resource-rich and country-specific and other resources such as the Media Literacy Council (https://www.betterinternet.sg; accessed on 24 August 2022) and Families for Life (https://www.familiesforlife.sg; accessed on 24 August 2022) and the American Academy of Pediatrics (https://www.healthychildren.org/English/family-life/Media/Pages/Tips-for-Parents-Digital-Age.aspx; accessed on 24 August 2022) that provide digital parenting tips and guidance for preschool children, children and adolescents, are some suggestions for positive action.

As families, children, and specific circumstances vary and differ across income and educational levels, and even then, can change over time (e.g., disruptions caused by COVID-19 pandemic), childcare and kindergartens in Singapore, working with the Early Childhood Development Agency in Singapore, and researchers can partner with parent-support groups to run parent mediation workshops. Research established that parent mediation or how parent engage or not engage with children around digital media can have positive effects on children use of media and can mitigate the negative effects [1]. Parents can be taught how to use these techniques via various avenues (e.g., workshops and seminars or via electronic factsheets). Accordingly, the three types of parent mediation styles are active mediation (parent communicates with children about media content, character action, and motivation), restrictive mediation (parent sets time limit and conditions for child media use), and social co-viewing (parent and child view or use media together but without active engagement with each other) [46].

There is a need for future research to investigate different parental mediation practices in the culture and context of Singapore, or in countries facing similar situations. Research should also be conducted to examine the effectiveness of digital parenting education programmes that inform parents on how to safeguard their child against cyber threats and the consequences of excessive media use [47].

## 5. Conclusions

Results show that parent digital use is moderately correlated with child digital media use. Children in the child-TQ were exposed to fixed screens at an earlier age than children in the child-BQ. Children in the child-TQ spent more time daily on digital media and physical play than children in the child-BQ. In spite of parent concerns about the possible negative impact of excessive digital media use among preschool children, the parental concerns did not translate sufficiently to action (i.e., enforcement of digital media use guidelines). Parents of children in the child-TQ appeared more persuaded about the merits of digital media use and were less likely to enforce digital media guidelines; they had lower than university level qualifications and came from less affluent households. No children in the child-TQ met *all* three WHO 24-hour integrated guidelines, with 12.1% meeting *none* of the same guidelines. Action-oriented and targeted approaches for parents identified as in need (lower-income households) and parent support groups that provide ‘point-of-decision’ practical strategies (parent mediation tips) may help ‘parents-at-risk’ of excessive daily digital media use to better manage the home environment for preschool children. Follow-up research on the effectiveness of community-based programmes and sustained parent mediation interventions are recommended. Comparative international research across countries using similar methodologies might also provide meaningful insights into an emergent problem in early childhood.

## Figures and Tables

**Table 1 ijerph-19-11508-t001:** Characteristics of top 25% and bottom 25% quartiles of child daily digital media use.

Parent-Reported SMALLQ ^a^	Top Quartile		Bottom Quartile		Mann–Whitney U/ Independent Samples *t*-Test/ Chi-Square Test of Independence
**Child characteristics**					
Age (years) 2 to 2.9 3 to 3.9 4 to 4.9	*n* (%) 90 (22.3) 162 (40.1) 152 (37.6)		*n* (%) 99 (24.6) 168 (41.7) 136 (33.7)		*X*^2^ (2) = 1.425 *p* = 0.49 *n* = 807
Age of first exposure to fixed screens (e.g., television) in months, *M (SD)*	17.1 (9.1)		18.9 (9.7)		*U* = 70695.5 ***p* = 0.005** *r* = −0.10 *n* = 799
Age of first exposure to mobile screens (e.g., mobile phones, tablets) in months, *M (SD)*	20.6 (9.8)		21.9 (10.5)		*U* = 68981.5 *p* = 0.110 *r* = −0.06 *n* = 769
Daily digital media use in hours, *M* (*SD*)	4.8 (2.2)		0.6 (0.3)		*U* = 0 ***p* = 0.001** *r* = 1.00 *n* = 807
Daily indoor and outdoor playtime in hours, *M* (*SD*)	3.6 (2.2)		2.7 (1.8)		*t* = −6.25 ***p* = 0.001** *d* = 0.44 *n* = 805
Total daily sleep (sum of day nap and night) in hours, *M* (*SD*)	10.5 (2.8)		10.6 (2.4)		*t* = 0.45 *p* = 0.651 *d* = −0.03 *n* = 805
Weekday sleep quality	*n*	*%*	*n*	*%*	
Unsatisfactory Below average Average Above average Excellent	1 13 133 132 98	0.3 3.4 35.3 35.0 26.0	7 15 121 150 94	1.8 3.9 31.3 38.7 24.3	*X*^2^ (4) = 6.31 *p* = 0.177 *n* = 764
Weekend sleep quality	*n*	*%*	*n*	*%*	
Unsatisfactory Below average Average Above average Excellent	1 9 106 145 115	0.3 2.4 28.2 38.5 30.6	3 11 117 146 110	0.8 2.9 30.2 37.7 28.4	*X*^2^ (4) = 1.70 *p* = 0.791 *n* = 763
Quality-of-life score (PEDsQL ^b^), *M* (*SD*)	77.9 (15.9)		80.2 (13.9)		*U* = 62877 *p* = 0.087 *r* = −0.07 *n* = 737
Meeting 24 h WHO guidelines ^c^	*n*	*%*	*n*	*%*	*X*^2^ (3) = 211.61 ***p* = 0.001** *n* = 807
None	49	12.1	0	0	
1	157	38.9	66	16.4	
2	198	49.0	212	52.6	
All 3	0	0	125	31.0	
**Parent characteristics**					
Composite digital media use in hours, *M* (*SD*)	9.0 (4.4)		5.1 (3.0)		*U* = 34494 ***p* = 0.001** *r* = 0.58 *n* = 807
Weekday in hours, *M* (*SD*)	10.5 (5.1)		6.5 (4.0)		*t* = −12.16 ***p* = 0.001** *d* = 0.86 *n* = 805
Weekend day in hours, *M* (*SD*)	7.6 (4.6)		3.6 (2.6)		*U* = 31621 ***p* = 0.001** *r* = 0.61 *n* = 807
Highest education level	*n*	*%*	*n*	*%*	*X*^2^ (1) = 31.47 ***p* = 0.001** *n* = 763
Below university University	226 151	59.9 40.1	153 233	39.6 60.4	
Monthly household income	*n*	*%*	*n*	*%*	
SGD 0– SGD 8000 SGD 8001 and above	295 79	78.9 21.1	242 136	64.0 36.0	*X*^2^ (1) = 20.32 ***p* = 0.001** *n* = 752

*Note*. When assumptions of normality are violated, Mann–Whitney U test was used as the non-parametric alternative. For categorical variables, chi-square test of independence was used. *d* = effect size reported in Cohen’s *d*. *r* = effect size reported in rank bi-serial correlation. ^a^ SMALLQ^®^ refers to Surveillance of Digital Media Habits in Early Childhood Questionnaire. ^b^ PEdsQL™ refers to Pediatric Quality of Life Inventory. ^c^ World Health Organization 24 h guidelines are: (i) less than 60 min of screen time; (ii) at least 180 min of physical activity; and (iii) 10–13 h of good quality sleep.

**Table 2 ijerph-19-11508-t002:** Parents’ Perceptions and Concerns of Child Digital Media Guidelines (top vs. bottom quartiles).

Parent Response	Top Quartile		Bottom Quartile		Chi-Square Test of Independence
	*n*	*%*	*n*	*%*	
Parental perceptions					
Improve child’s knowledge and skills					X^2^ (1) = 51.18
Not very important	160	39.6	261	64.8	***p* = 0.001**
Very important	244	60.4	142	35.2	*n* = 807
Keep child entertained					X^2^ (1) = 29.34
Not very important	232	57.4	304	75.4	***p* = 0.001**
Very important	172	42.6	99	24.6	*n* = 807
Communication					X^2^ (1) = 6.10
Not very important	259	64.1	291	72.2	***p* = 0.014**
Very important	145	35.9	112	27.8	*n* = 807
Keep child occupied					X^2^ (1) = 21.17
Not very important	238	58.9	299	74.2	***p* = 0.001**
Very important	166	41.1	104	25.8	*n* = 807
Distract or divert child’s attention					X^2^ (1) = 14.58
Not very important	274	67.8	321	79.7	***p* = 0.001**
Very important	130	32.2	82	20.3	*n* = 807
Put child to sleep					X^2^ (1) = 0.45
Not very important	339	83.9	345	85.6	*p* = 0.502
Very important	65	16.1	58	14.4	*n* = 807
Parental concerns					
Poor sleep					X^2^ (1) = 2.98
Not very concerned	208	51.5	183	45.4	*p* = 0.084
Very concerned	196	48.5	220	54.6	*n* = 807
Poor eyesight					X^2^ (1) = 0.01
Not very concerned	108	26.7	109	27.0	*p* = 0.92
Very concerned	296	73.3	294	73.0	*n* = 807
Lack of physical exercise and activity					X^2^ (1) = 1.08
Not very concerned	180	44.6	165	40.9	*p* = 0.30
Very concerned	224	55.4	238	59.1	*n* = 807
Exposure to inappropriate content					X^2^ (1) = 0.29
Not very concerned	110	27.2	103	25.6	*p* = 0.591
Very concerned	294	72.8	300	74.4	*n* = 807
Addiction					X^2^ (1) = 0.12
Not very concerned	96	23.8	100	24.8	*p* = 0.728
Very concerned	308	76.2	303	75.2	*n* = 807
Lack of parent–child interaction					X^2^ (1) = 0.002
Not very concerned	140	34.7	139	34.5	*p* = 0.961
Very concerned	264	65.3	264	65.5	*n* = 807

*Note*. Survey responses indicating ‘not important’, ‘of little importance’, or ‘moderately important’ in SMALLQ^®^ were classified as not very important, and responses ‘indicating important’ or ‘very important’ were classified as very important. Survey responses indicating ‘not concerned’, ‘minimally concerned’, and ‘somewhat concerned’ in SMALLQ^®^ were classified as not very concerned, and responses indicating ‘concerned’ and ‘seriously concerned’ were classified as very concerned.

**Table 3 ijerph-19-11508-t003:** Parents’ Knowledge and Enforcement of Child Digital Media Guidelines (top vs. bottom quartiles).

Parent Response to Child Digital Media Guidelines ^a^	Top Quartile		Bottom Quartile		
	*n*	*%*	*n*	*%*	*p*
Limit screen time to no more than 1 h per day for children aged 2–5 years					
Not aware	82	20.3	34	8.4	
Not aware but practising	53	13.1	45	11.2	X^2^ (3) =134.59
Aware but not practising	139	34.4	39	9.7	***p* = 0.001**
Aware and practising	130	32.2	285	70.7	*n* = 807
Introduce only high-quality educational content or programmes for children					
Not aware	97	24.0	74	18.4	
Not aware but practising	62	15.4	50	12.4	X^2^ (1) = 17.93
Aware but not practising	64	15.8	40	9.9	***p* = 0.001**
Aware and practising	181	44.8	239	59.3	*n* = 807
Co-watch or co-play digital media with children					
Not aware	49	12.1	37	9.2	
Not aware but practising	67	16.6	41	10.2	X^2^ (1) = 10.35
Aware but not practising	45	11.1	55	13.6	***p* = 0.016**
Aware and practising	243	60.2	270	67.0	*n* = 807

^a^ Child digital media guidelines were adapted and truncated from the American Academy of Pediatrics.

## Data Availability

The data associated with this study were published in NIE data repository as per the requirements of the corresponding author’s university.

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
