# Peer review of "Associations between Parent Attitudes and on- and off-Screen Behaviours of Preschool Children in Singapore"

_ijerph, 2022, doi:10.3390/ijerph191811508_

Round 1

Reviewer 1 Report

Introduction commments.

Within the introduction is would be helpful to expand out more the "better outcomes", what are these? You just say that there are these, but a bit more specifics would help this section. "Researchers posit that parental mediation of screen media is associated with better outcomes for children"

To offer some global comparison it would be good please to add context / comparison of the levels of mobile use in Signapore compared to elsewhere, is this significantly higher, the same etc. this would help for international readers to compare to their home context.

Within the introduction it would be interesting to offer some more details about the change in media or what is that 'hooks' preschool children into almost being addicted to this media, what make them want to have it? 

Can you expand how the WHO guidelines have impacted the Signapore national guidelines for screen time, and in the context of when you undertook the study in 2019, what was the impact of the WHO new guidelines, or is this a limitation of the timing of your study, if it is considered a limitation, please note this and highlight how you might get different results now there has been 3 years of knowing more about these guidelines.

Within the study purpose, can you add specifics please as to how the characteristics of the top 25% (child-TQ) versus bottom 25% (child-BQ) quartiles of daily child digital media use were decided, it is not fully clear reading the purpose, to allow others to fully replicate your study if they wish too. This would help understand if it was top and bottom for each age group, or if this was overall.

In Materials and Methods section, please add in more demographical details, e.g. what percentage of parents were recruited, how many did not return? You discuss your removal processes, but how many questionnaires / responses did you lose, what is your final total of parents? You come to the precise number later, but in the purpose you target number would be good to add please.

I am impressed by all your face validity and reliability elements that you undertook of the questionnaires - good job.

In section 2.4.1. please note you have a typo of an extra full stop after the 1. Also could you add a reference to support your use of self reporting, especially for this age group, this would enhance and support this section.

Please add into 3 the precise characteristics of the average age of the children, boys and girls, plus standard deviations, plus add in the number of 2 year olds, 3 year olds, 4, year olds, 5 years olds.
This extra comparision, may need extra data analysis or addition of data, but as the impact of growth and development is key within the 2 - 5 year age bracket, more detail is needed please. Could you add in if there is a difference in time spent in preschool according to age. In the discussion you comment about time spent from 7 hours to 12 hours, but you don't link this to age, so it is unclear as to how this may have impacted your results. 

Within the results section, please can you format the tables so the words are in line with the numbers, the table are hard to read to follow which numbers are linked to which.

In the discussion section - please expand out more the "better outcomes" as per introduction comment. Again it isn't clear what better outcomes are referring too.

Fascinating result re the parents and weekend time, vs child weekend time and the notion of 'digital nanny'.

In 4.6 can I check that you don't have a typo. Did you really mean 550% or 50%? I can't see 550% in you results table.

Please add into the limitations the timeline of when you undertook the research and the introduction of the WHO guidelines, this may have impacted the lack of parental engagement in the guidelines, this is important to note.

Overall a really fascinating manuscript. Thank you for allowing me the opportunity to review. I hope my comments are helpful and I would recommend if possible adding in the additional data analysis per age as this is impacted by the number of hours in preschool and therefore the opportunities to access screen time.

Reviewer 2 Report

In this article, the authors evaluated the association between parental attitudes and on- and off-screen behaviors of preschool children in Singapore.

Introduction: reflects, in various sections, all the relevant information on the topic to be discussed.

Material and methods: all parents who conducted the survey belonged to the same social environment? Please clarify.

Are the surveys used validated? Please clarify.

What were the inclusion and exclusion criteria of the study? children with previous vision problems were taken into account?

Discussion: Do you know what activity a 2-year-old child does with a digital device?

Children in the study used digital devices at school?

Conclusions: the first part of the conclusions are results rather than conclusions.

Author Response

Please see the attachment- authors' responses to reviewer 2.

Reviewer 3 Report

The research topic is very current and relevant.

Overall, the language use is very good.

A more critical synthesis of the literature and the potential construction of a research agenda will also improve the significance and value of the manuscript.

Author(s) need to mention ethical issues for their study and the relations between science and society. I propose to add the following reference:

Petousi, V., & Sifaki, E. (2020). Contextualizing harm in the framework of research misconduct. Findings from discourse analysis of scientific publications, International Journal of Sustainable Development, 23(3/4), 149-174, DOI: 10.1504/IJSD.2020.10037655

https://www.inderscienceonline.com/doi/abs/10.1504/IJSD.2020.115206

Authors need to strengthen their theoretical framework. I propose the following articles:

Vaiopoulou, J., Papadakis, S., Sifaki, E., Stamovlasis, D., Kalogiannakis, M. (2021). Parents’ Perceptions of Educational Apps Use for Kindergarten Children: Development and Validation of a New Instrument (PEAU-p) and Exploration of Parents’ Profiles, Behavioral Sciences, 11(6), 82, https://doi.org/10.3390/bs11060082

Konca, A. S., & Tantekin Erden, F. (2021). Young Children’s Social Interactions with Parents during Digital Activities at Home. Child Indicators Research14(4), 1365-1385.

Papadakis, St., Zaranis, N. & Kalogiannakis, M. (2019). Parental involvement and attitudes towards young Greek children’s mobile usage. International Journal of Child-Computer Interaction, 22(2019) 100144, https://doi.org/10.1016/j.ijcci.2019.100144

I wish you the best of luck with the revisions of your manuscript.

Author Response

Please see the attachment- authors' responses to reviewer 3.

Reviewer 4 Report

Thank you very much for the chance of reviewing this interesting manuscript. The behaviours of parents certainly influence their children a lot and the addiction to digital media use is one of the concerns among people of different age ranges in society. Therefore, the findings of this article can be an important reference for governments and related practitioners to formulate some strategies to improve this concern. Overall, the study is good, but minor issues listed below needed to be addressed. 

1.     The introduction of this study is clear and detailed as the related issues were systematically demonstrated and explained. Also, the purposes were clearly stated. However, only “These data provide ‘just-in-time’ and ‘current-state’ evidence to inform parents, policies and processes that enhance the quality of early childhood education at home and in partnership with school” was stated to indicate the significance of the study. I do believe that the findings of this study can contribute more than this and thus try to be more specific and detailed about the importance of this research.

2.     It is important to explain why to select the participants were aged 2 – 5 years old. Why not aged 2-3 only or aged 2-6 years old? Whether children aged 2-5 are more easily influenced by their parent's behaviours. Based on what rationales?

3.     Why would you choose the top and bottom quartiles as one of the data analysis methods? You may indicate the reason why using this approach.

Author Response

Please see the attachment- authors' responses to reviewer 4.

Round 2

Reviewer 1 Report

Thank you for completing all the suggested recommendations, amendments and adding additional explanations. I felt this has really enhanced your manuscript and it is a really fascinating article.

Well done.